# Are Δ^9^-Tetrahydrocannabinol and Its Major Metabolites Substrates or Inhibitors of Placental or Human Hepatic Drug Solute-Carrier Transporters?

**DOI:** 10.3390/ijms252212036

**Published:** 2024-11-09

**Authors:** Xin Chen, Zsuzsanna Gáborik, Qingcheng Mao, Jashvant D. Unadkat

**Affiliations:** 1Department of Pharmaceutics, School of Pharmacy, University of Washington, Seattle, WA 98195, USA; cx2436cx@uw.edu (X.C.); qmao@uw.edu (Q.M.); 2SOLVO Biotechnology, Charles River Laboratories Hungary, Irinyi József u. 4-20, 1117 Budapest, Hungary; zsuzsanna.gaborik@crl.com

**Keywords:** THC, transporters, solute carrier, placenta, liver, pregnancy

## Abstract

Δ^9^-Tetrahydrocannabinol (THC) is the primary psychoactive component of cannabis which is being increasingly consumed by pregnant people. In humans, THC is sequentially metabolized in the liver to its circulating metabolites 11-hydroxy-THC (11-OH-THC, psychoactive) and 11-*nor*-9-carboxy-THC (THC-COOH, non-psychoactive). Human and macaque data show that fetal exposure to THC is considerably lower than the corresponding maternal exposure. Through perfused human placenta studies, we showed that this is due to the active efflux of THC (fetal-to-maternal) by a placental transporter(s) other than P-glycoprotein or breast cancer resistance protein. The identity of this placental transporter(s) as well as whether THC or its metabolites are substrates or inhibitors of hepatic solute carrier transporters is unknown. Therefore, we investigated whether 5 μM THC, 0.3 μM 11-OH-THC, and 2.5 μM THC-COOH are substrates and/or inhibitors of placental or hepatic solute carrier transporters at their pharmacologically relevant concentrations. Using HEK cells overexpressing human OATP1B1, OATP1B3, OATP2B1, OCT1, OCT3, OAT2, OAT4, or NTCP, and prototypic substrates/inhibitors of these transporters, we found that THC and THC-COOH were substrates but not inhibitors of OCT1. THC-COOH was a weak substrate of OCT3 and a weak inhibitor of OAT4. THC, 11-OH-THC, and THC-COOH were found not to be substrates/inhibitors of the remaining transporters investigated.

## 1. Introduction

The use of cannabis by pregnant people in the United States is increasing [1,2], raising concerns about the potential fetal and neurodevelopmental toxicity [3,4] of (−)-Δ^9^-tetrahydrocannabinol (THC), the primary psychoactive constituent of cannabis. To determine this toxicity, it is important to determine the mechanism and extent of placental transfer of THC and its psychoactive circulating metabolite 11-OH-THC.

In both humans and non-human primates, THC is effluxed in the fetal-to-maternal direction, resulting in fetal circulatory concentrations that are less than the corresponding maternal concentrations [5,6]. These data suggest the involvement of efflux transporters at the blood–placenta barrier, particularly P-glycoprotein (P-gp) and/or breast cancer resistance protein (BCRP), which are highly expressed in the apical membrane of the syncytiotrophoblast [7]. This hypothesis is supported by some studies in the P-gp knock-out (KO) mice [8], but not by those conducted in our laboratory [9,10]. Using both P-gp- or BCRP-overexpressing cells or vesicles, or pregnant P-gp or BCRP KO mice, we found that THC is not a substrate or inhibitor of P-gp or BCRP at pharmacologically relevant concentrations [9,10]. Moreover, in our perfused human placenta studies, the THC unbound clearance (CL) in the fetal-to-maternal direction (normalized to the unbound CL of the passive diffusion marker antipyrine) was significantly greater than its corresponding maternal-to-fetal direction, indicating active efflux of THC in the maternal-to-fetal direction. In addition, this active efflux was not inhibitable by a pan-P-gp/BCRP inhibitor (valspodar) [6]. In these perfused placental studies, 11-OH-THC and THC-COOH were found to cross the placenta passively [6].

The above findings raise an interesting question: which transporter(s) is (are) responsible for the fetal-to-maternal efflux of THC? Studies to predict and verify the observed fetal umbilical vein concentrations in our human studies through physiologically based pharmacokinetic (PBPK) modeling suggest that the transporter(s) is (are) more likely to be localized in the basal membrane of the syncytiotrophoblast. We hypothesized that this transporter could be one of the solute carrier (SLC) uptake transporters, such as the organic anion-transporting polypeptide (OATP2B1), the organic cation transporter (OCT3), or the organic anion transporter (OAT4) (Figure 1A) [11,12]. That is, THC could be actively transported from the fetal circulation into the syncytiotrophoblast by one of these transporters and then THC could reach the maternal circulation by diffusion across the apical membrane of the syncytiotrophoblast. Therefore, here, we investigated if THC or its major circulatory metabolites are substrates or inhibitors of these basolateral placental transporters at their pharmacologically relevant concentrations.

In humans, THC is primarily metabolized in the liver [via cytochrome P450 (CYP) 2C9] to its psychoactive circulatory metabolite, 11-OH-THC, which is further metabolized (via CYP2C9 and 3A) to the non-psychoactive circulatory metabolite 11-*nor*-9-carboxy-THC (THC-COOH) [13,14]. After intravenous dosing of THC (0.5 mg), 41–45% of the dose is excreted in the feces 72 h after administration [15]. It is possible that hepatic transporters may be important in the hepatic uptake of THC or its metabolites and their subsequent excretion in the bile or urine. If they are, this has potential implications for drug–drug interactions, either as object drugs or as perpetrators when present in the circulation at their upper range of their pharmacological concentrations (5 μM THC, 0.3 μM 11-OH-THC, or 2.5 μM THC-COOH, as justified by our previous publication [9]). Therefore, we also investigated if THC and its major circulatory metabolites are substrates or inhibitors of hepatic SLC transporters such as OATP1B1/1B3/2B1, OCT1, and sodium taurocholate co-transporting polypeptide (NTCP) (Figure 1B) [16].

## 2. Results

### 2.1. Uptake of Cannabinoids by the Basal Syncytiotrophoblast Transporters

Although day-to-day variability was observed, the uptake of THC or 11-OH-THC by HEK-OATP2B1, OCT3, or OAT4 cells was not significantly different in the presence of the respective transporter inhibitor compared with that in the absence of the inhibitor (DMSO only) (Figure 2). Likewise, cellular accumulation of THC-COOH in these cells was not affected by the respective transporter inhibitor except in HEK-OCT3 cells where it was slightly but significantly decreased in the presence of 100 μM corticosterone (Figure 2). Interestingly, bromsulphthalein significantly, but modestly, stimulated (rather than inhibited) the uptake of THC-COOH into HEK cells expressing OAT4 (Figure 2).

### 2.2. Inhibition of the Basal Syncytiotrophoblast Transporters by the Cannabinoids

To determine if THC and its major metabolites are inhibitors of these placental transporters at their pharmacologically relevant concentrations, the uptake of probe substrates of OATP2B1/OCT3/OAT4 (25 nM [^3^H]-estrone-3-sulfate for OATP2B1 and OAT4, 1.67–2 μM [^14^C]-metformin for OCT3) was determined in the presence or absence of the cannabinoids or known inhibitors of the respective transporter (positive control). Although THC did not inhibit the uptake of the prototypic substrates by the tested placental transporters, THC-COOH slightly, but significantly, inhibited OAT4-mediated uptake of estrone-3-sulfate, while 11-OH-THC and THC-COOH modestly increased (43% and 19%, respectively) OATP2B1-mediated uptake of estrone-3-sulfate (Figure 3).

### 2.3. Uptake of Cannabinoids by the Sinusoidal Hepatic Transporters

OCT1-mediated uptake of THC and THC-COOH (but not of 11-OH-THC) was decreased by 51% and 38%, respectively, in the presence of the OCT1 inhibitor quinidine (100 µM). In contrast, the uptake of the cannabinoids by OATP1B1/OATP1B3/OAT2/NTCP was not affected by their respective inhibitors. 11-OH-THC was not found to be a substrate of OATP1B1, OATP1B3, OCT1, OAT2, or NTCP (Figure 4).

### 2.4. Inhibition of the Sinusoidal Hepatic Transporters by the Cannabinoids

At their pharmacologically relevant concentrations, none of the cannabinoids were found to inhibit the uptake of substrates of OATP1B1, OATP1B3, OCT1, OAT2, or NTCP (Figure 5).

## 3. Discussion

In this study, we systemically evaluated whether THC and its major metabolites, 11-OH-THC and THC-COOH, at their pharmacologically relevant concentrations are substrates or inhibitors of numerous placental and hepatic SLC transporters. The impetus for this study arose from our previous findings that THC (but not 11-OH-THC or THC-COOH) is transported in the fetal-to-maternal direction in term perfused human placentas [6]. An obvious explanation is that THC is transported by either P-gp and/or BCRP, two efflux transporters highly expressed in the apical membrane of the syncytiotrophoblast (Figure 1). However, in our perfused human placenta studies, the fetal-to-maternal THC transport was not inhibitable by valspodar, an inhibitor of P-gp, BCRP, and multidrug resistance-associated protein 4 (MRP4). In addition, in studies in cells overexpressing human P-gp or BCRP or in mice where P-gp and/or Bcrp was knocked out, THC was not transported by either P-gp or Bcrp [9,10]. An alternative explanation is that THC is effluxed in the perfused human placenta by MRP2/3 expressed in the apical membrane of the syncytiotrophoblast. However, through PBPK modeling and simulations together with the observed THC concentrations in the fetal circulation and tissues, we concluded that THC is most likely effluxed in the fetal-to-maternal direction via uptake transporters in the basal membrane of the syncytiotrophoblast [17]. Thus, based on our previous quantitative targeted proteomics data, we chose to investigate OATP2B1, OCT3, and OAT4 as potential candidates as they are highly expressed in the basal membrane of the syncytiotrophoblast [12] (Figure 1). Although OAT10 is also expressed there [18] (not quantified in our proteomics study), we did not have access to a cell line overexpressing this transporter. We deliberately utilized cells that overexpress human SLCs (such as HEK293 cells) to allow for the maximum sensitivity to identify the transporter(s) involved in the placental efflux of THC. Endogenous transporters in our HEK cell lines were not a confounding factor as the uptake of cannabinoids or the transporter prototypic substrates in the presence and absence of selective inhibitors of the transporters in the non-transfected HEK cells was found not to be significantly different.

We found that THC and 11-OH-THC are not substrates of OATP2B1, OCT3, or OAT4 (Figure 2, Appendix A). Therefore, the identity of the transporters mediating fetal-to-maternal efflux of THC remain unknown. At the same time, we cannot completely discount efflux transporters in the apical membrane of the syncytiotrophoblast other than P-gp, BCRP, or MRP4. Possible candidates are MRP2 and MRP3 based on immunohistochemistry, colocalization, and proteomics data [11,12,19]. We were unable to investigate these transporters as cell lines expressing these transporters to conduct efflux studies in Transwells were not available to us. Experiments using Transwells (rather than using membrane vesicles) is best suited to study the transport of highly lipophilic compounds such as THC (pK_a_ = 10.6, Log P = 6.97), which binds avidly to labware. Moreover, the majority of the reported substrates of MRP2/3 are hydrophilic compounds [20].

Surprisingly, THC-COOH (an anion, pK_a_ = 4.2, Log P = 5.24) was found to be a weak substrate of OCT3 and therefore OCT3 could potentially limit fetal exposure to THC-COOH provided that OCT3 transports drugs in the fetal-to-maternal direction. There is evidence to suggest that OCT3 transport can be bidirectional [21]. In *Oct3*^−/−^ pregnant mice, the fetal-to-maternal area under the curve (AUC_0–∞_) ratio exposure to metformin, an OCT3 substrate, was decreased by 44% [21]. Irrespective of the directionality of OCT3 transport, our perfused human placenta studies indicate that THC-COOH passively crosses the placenta [6]. These data indicate that any in vivo (or ex vivo) transport of THC-COOH by placental OCT3 is likely negligible. OCT3 is highly expressed in the human placenta but not in the human liver and kidneys where OCT1 and OCT2 are, respectively, highly expressed [22]. Thus, it is likely to play a smaller role than OCT1 in the distribution of THC-COOH in the liver (see below for further discussion). Prompted by our findings, it would be interesting to determine if THC-COOH is a substrate of OCT2.

None of the cannabinoids were found to be inhibitors of the investigated placental SLC transporters except for THC-COOH which weakly inhibited OAT4 (Figure 3, Appendix A). Given the high plasma protein binding of THC-COOH (96.35% in mice) [10] and its estimated in vivo maximal fetal plasma concentrations (~125 nM after maternal oral administration of 14.8 mg THC [23,24] and assuming it crosses the placenta by passive diffusion, it is unlikely to result in any in vivo inhibition of placental OAT4. Interestingly, 11-OH-THC and THC-COOH slightly stimulated OATP2B1-mediated uptake of estrone-3-sulfate, suggesting that they allosterically modulate OATP2B1 activity.

The blood CL of THC in humans after intravenous or inhalation administration is blood-flow limited [25] while its oral CL is dependent on its intrinsic hepatic CL. However, it is unknown whether THC utilizes transporters to gain entry into the hepatocytes. If it does, drug–drug interactions or genetic polymorphism of the transporters may affect its blood CL (especially after oral administration). Therefore, we investigated whether THC and its major metabolites are substrates or inhibitors of sinusoidal transporters namely, OATP1B1, OATP1B3, OATP2B1, OCT1, OAT2, and NTCP.

Except for OCT1, none of the cannabinoids were substrates or inhibitors of the investigated transporters. We found that THC and THC-COOH are OCT1 substrates (Figure 4, Appendix A) but not OCT1 inhibitors at their pharmacologically relevant concentrations (Figure 5, Appendix A). The finding that OCT1 can transport THC was surprising because THC is an anion at physiological pH (pK_a_ = 10.6; [26]). Anions transported by OCTs are usually zwitterions like creatinine and cimetidine [27,28]. Based on these data, THC and THC-COOH are unlikely to be perpetrators of OCT1–drug interactions but could be objects of OCT1-related cannabinoid-drug interactions provided that their fraction transported in vivo by OCT1 is significant.

Our data have some limitations. First, there was a moderate degree of inter-day variability in transporter activity. However, this does not detract from the conclusions drawn. We were not able to evaluate all the major drug transporters expressed in the placenta (e.g., MRPs) due to the lack of availability of cells that express these transporters and can form tight junctions (e.g., Madin–Darby canine kidney cells).

In summary, we found that THC is not a substrate of OATP2B1, OCT3, or OAT4. Therefore, the identity of the transporters mediating fetal-to-maternal efflux of THC remain unknown. Perhaps THC is a substrate of one of the many transporters that transport endogenous compounds (e.g., nutrients) and are distinct from the transporters investigated here. Further studies are needed to identify this transporter. Surprisingly, we found that THC and THC-COOH are substates of the hepatic transporter OCT1. Thus, they could be objects of OCT1-related cannabinoid-drug interactions provided that their fraction that is transported in vivo by OCT1 is significant.

## 4. Materials and Methods

### 4.1. Materials

(−)-Δ^9^-THC (50 mg/mL) was purchased from Cayman Chemicals (Ann Arbor, MI, USA). (±)11-OH-THC (1 mg/mL), (±)-11-*nor*-9-carboxy-THC (THC-COOH) (1 mg/mL), (−)-Δ^9^-THC-D_3_ (1 mg/mL), (±)11-OH-THC-D_3_ (1 mg/mL), and (±)-11-*nor*-9-carboxy-Δ9-THC-D_3_ (1 mg/mL) in methanol were from Cerilliant Corporation (Round Rock, TX, USA). Bromsulphthalein, bulevirtide trifluoroacetate, corticosterone, erlotinib, ketoprofen, quinidine, and rifampin were from Sigma-Aldrich (St. Louis, MO, USA). Regarding radiolabeled compounds, 1 mCi/mL (25 Ci/mmol) [^3^H]-rosuvastatin, 1 mCi/mL (40 Ci/mmol) [^3^H]-estrone-3-sulfate, 1 mCi/mL (25 Ci/mmol) [^3^H]-cGMP, and 1 mCi/mL (20 Ci/mmol) [^3^H]-taurocholic acid were from American Radiolabeled Chemicals (St. Louis, MO, USA), while 0.1 mCi/mL (110 mCi/mmol) [^14^C]-metformin was from Moravek Biochemicals, Inc. (Brea, CA, USA). Hank’s balanced salt solution (HBSS), 0.25% trypsin–EDTA, GlutaMAX, fetal bovine serum (FBS), Dulbecco’s modified eagle medium (DMEM) (4.5 g/L glucose and 1.0 g/L glucose), dimethyl sulfoxide (DMSO), acetonitrile (liquid chromatography–mass spectrometry grade), acetic acid (liquid chromatography–mass spectrometry grade), and SureSTART™ polypropylene insert (Catalog #6EME03CPPSP) were from ThermoFisher Scientific (Hampton, NH, USA). Poly-D-lysine-coated 48-well cell culture plates were from Corning (Corning, NY, USA). Low-binding microcentrifuge tubes were from Genesee Scientific (San Diego, CA, USA). Milli-Q water was used in all preparations. High-Density Polyethylene MiniVials (7 mL, catalog #125,500) were from RPI Research Products International (Mount Prospect, IL, USA). All other chemicals and reagents were obtained commercially at the highest quality available.

### 4.2. Methods

#### 4.2.1. Cell Culture

HEK-OATP2B1, HEK-OCT1, HEK-OCT3, HEK-OAT4, and HEK-NTCP cells were generously provided by SOLVO Biotechnology (Szeged, Hungary). HEK-OATP1B1 and HEK-OATP1B3 cells were generously provided by Gilead Sciences Inc. (Foster City, CA, USA). HEK-OAT2 cells were generously provided by Pfizer Inc. (Cambridge, MA, USA). All cells were preserved in liquid nitrogen. The passage number of all the cells that were used was no greater than 10. Except for HEK-OATP1B1 and HEK-OATP1B3 cells, all cells were cultured in high-glucose (4.5 g/L) DMEM supplemented with 10% FBS, 1% GlutaMAX, and 1% penicillin–streptomycin. HEK-OATP1B1 and OATP1B3 cells were cultured in high-glucose (4.5 g/L) DMEM supplemented with 10% FBS, 1% penicillin–streptomycin, 25 mM HEPES, 0.1 mM MEM non-essential amino acid solution, 600 μg/mL geneticin, and 10 μg/mL blasticidin. All cells were maintained in a humidified incubator at 37 °C in 5% CO_2_ with 95% humidity. When cells reached ~90% confluency, after washing with HBSS and trypsinization, they were passaged into a new flask, and then seeded into 48-well plates at a density of 20,000 cells per well for transport assays.

#### 4.2.2. Cannabinoid Uptake by SLC Transporters

When confluent in the 48-well plate, the cells were rinsed twice with 0.5 mL 37 °C HBSS and preincubated with 0.5 mL HBSS at 37 °C for 15 min. In order to ensure that the contents of the transport assays were at 37 °C, pilot studies showed that the setting of the water bath or hot plate needed to be at 42 °C. After aspiration of the HBSS, the cells were incubated with 5 μM THC, 0.3 μM 11-OH-THC, or 2.5 μM THC-COOH with or without the transporter inhibitor (or inhibitory condition) in HBSS (0.2 mL) containing DMSO (final concentration < 0.2% *v*/*v*) at 37 °C for 15 s (preliminary studies showed that this time was within the linear uptake range). After aspirating the HBSS, the cells were immediately washed three times with 0.5 mL of ice-cold HBSS. Then, cells were lysed with 200 μL of 80% acetonitrile containing 100 nM THC-D3, 100 nM 11-OH-THC-D3, or 100 nM THC-COOH-D3. All cell lysates were vortexed for 15 s and centrifuged at 19,083× *g* for 15 min at 4 °C. The supernatant (100 μL) was transferred to a disposable clean SureSTART™ polypropylene insert and stored at −20 °C until analysis by liquid chromatography–tandem mass spectrometry (LC-MS/MS).

#### 4.2.3. LC-MS/MS Analysis

On the day of analysis, 10 μL of sample was injected onto LC-MS/MS for analysis using a Xevo TQ-XS Triple Quadrupole Mass Spectrometer (Waters, Milford, MA, USA) coupled with an Acquity ultra-performance liquid chromatography system (Waters, Milford, MA, USA). The samples were eluted on an Acquity ultra-performance liquid chromatography BEH C_18_ column (130 Å, 1.7 µm, 2.1 mm × 50 mm) attached to the AccQ Tag Ultra C_18_ VanGuard Pre-column (100 Å, 1.7 µm, 2.1 mm × 5 mm). For other analytical conditions, please see Appendix A. All samples were quantified and analyzed by MassLynx™ V4.2. Given that the corresponding deuterated cannabinoid was employed as an internal standard, a single-point calibration method was utilized to quantify the molar concentration of the analytes, assuming that the analyte and the internal standard produced equivalent signal intensities.

#### 4.2.4. Inhibition of SLC Transporters by the Cannabinoids

When confluent in 48-well plate, the cells were processed as described above. After aspirating the HBSS, the cells were incubated with the radiolabeled transporter substrate, transport inhibitor (see Table 1 for concentrations), 5 μM THC, 0.3 μM 11-OH-THC, or 2.5 μM THC-COOH in 0.2 mL of HBSS containing DMSO (final concentrations < 0.2% *v*/*v*) at 37 °C for 2 min for substrates labeled with [^3^H] or for 10 min for substrates labeled with [^14^C] (preliminary studies showed that these times were within the linear uptake range). Uptake was quenched by aspirating the buffer and immediately washing the cells three times with ice-cold HBSS. Then, the cells were lysed with 80% acetonitrile and 100 μL of the cell lysate was added to 5 mL of Ecoscint™ (National Diagnostics, Atlanta, GA, USA) in a High-Density Polyethylene MiniVial and capped. After 15 s of mixing on a vortex, the radioactivity (corrected for background radioactivity) was quantified using a scintillation counter (PerkinElmer Tri-Carb 3110TK, GMI Laboratory Solutions, Ramsey, MN, USA).

#### 4.2.5. Statistical Analysis

Three independent experiments were performed, each in triplicate, for all the uptake and inhibition assays. Data are reported as the mean of each experiment and were analyzed using the paired *t*-test. *p*-values < 0.05 were considered statistically significant. All analyses were performed using GraphPad Prism 10 (La Jolla, CA, USA).

## 5. Conclusions

In conclusion, our data show that the hepatic clearance of THC and THC-COOH could be objects of hepatic OCT1-related cannabinoid-drug interactions provided that their fraction transported in vivo by OCT1 is significant. In addition, the basolateral placental transporters OATP2B1, OCT3, and OAT4 cannot explain the observed reduced fetal exposure to THC in humans. Additional studies (beyond the scope of this paper) are needed to identify this transporter.

## Figures and Tables

**Figure 1 ijms-25-12036-f001:**
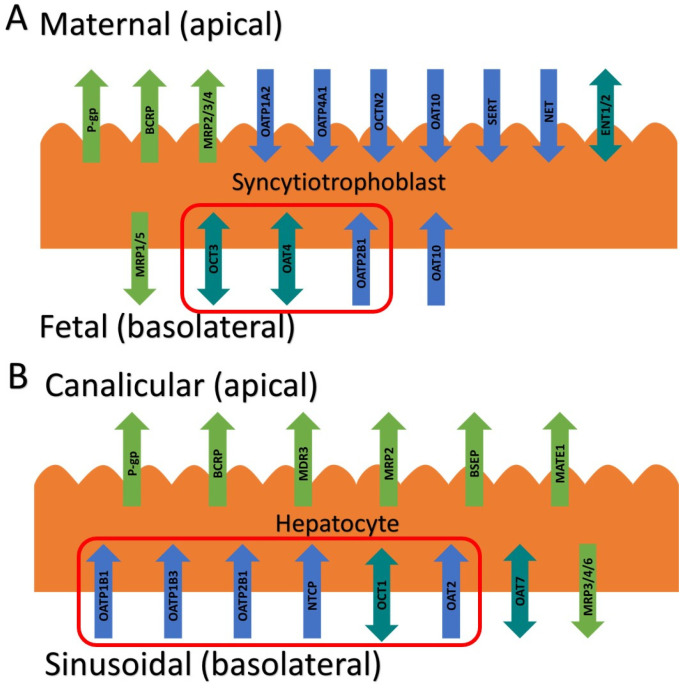
Placental (**A**) and hepatic (**B**) transporters with their localization and directionality of transport. Transporters in red boxes are those that were investigated. BCRP, breast cancer resistance protein; BSEP, bile salt export pump; ENT, equilibrative nucleoside transporter; MATE, multidrug and toxin extrusion protein; MDR, multidrug resistance protein; MRP, multidrug resistance-associated protein; NET, norepinephrine transporter; NTCP, sodium taurocholate cotransporter protein; OAT, organic anion transporter; OATP, organic anion-transporting peptide; OCT, organic cation transporter; OCTN, organic cation/carnitine transporter; P-gp, P-glycoprotein; SERT, serotonin transporter.

**Figure 2 ijms-25-12036-f002:**
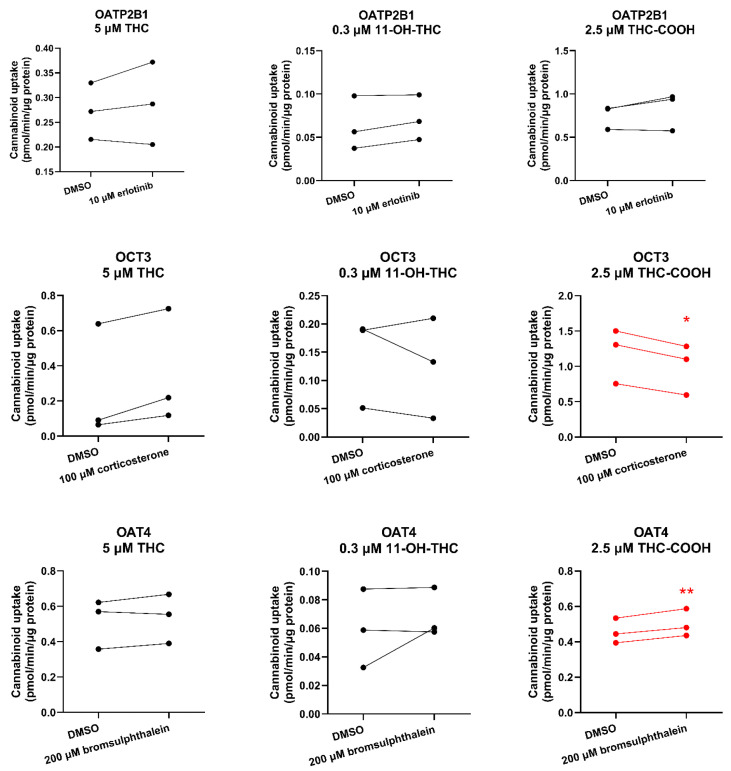
Cannabinoids as substrates of placental transporters. Uptake (over 15 s) of 5 μM THC, 0.3 μM 11-OH-THC, and 2.5 μM THC-COOH by HEK cells overexpressing human placental transporters in the absence or presence of the prototypic inhibitors of the respective transporter (10 μM erlotinib for OATP2B1, 100 μM corticosterone for OCT3, 200 μM bromsulphthalein for OAT4). Data on the *y*-axis are the uptake rate of the cannabinoid normalized to protein concentration. Data are from three independent experiments, each conducted in triplicate. Technical variability within each independent experiment is shown in Appendix A. Statistically significant differences in cannabinoid uptake in the presence or absence of the prototypic inhibitor (marked in red) were determined using the paired *t*-test. * *p* < 0.05; ** *p* < 0.01.

**Figure 3 ijms-25-12036-f003:**
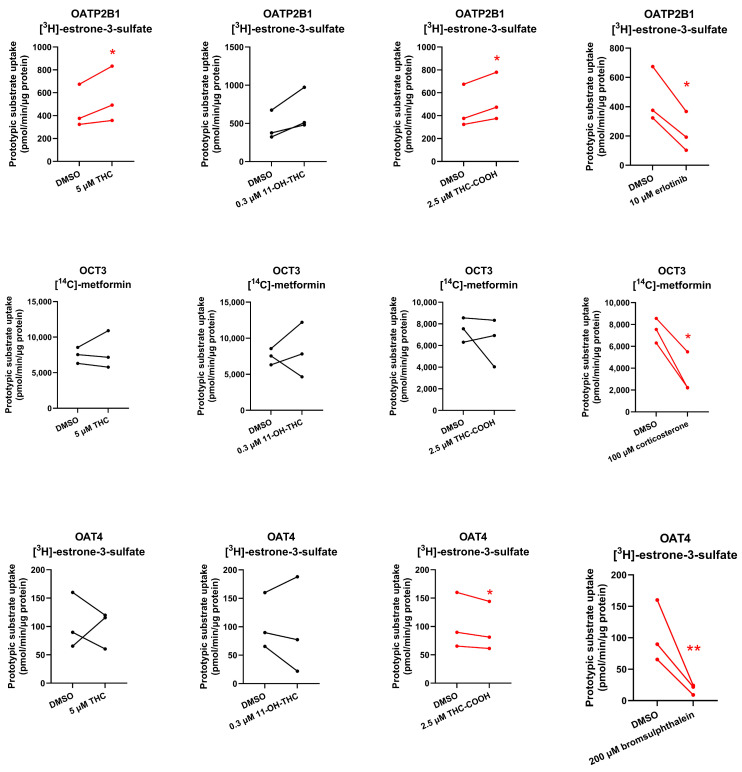
Cannabinoids as inhibitors of placental transporters. Rate of uptake (over 2 min for OATP2B1/OAT4 and 10 min for OCT3) of prototypic substrates of the placental transporters OATP2B1/OAT4 ([^3^H]-estrone-3-sulfate) or OCT3 ([^14^C]-metformin) in the presence or absence of 5 μM THC, 0.3 μM 11-OH-THC, 2.5 μM THC-COOH, or their prototypic selective inhibitors (10 μM erlotinib for OATP2B1, 100 μM corticosterone for OCT3, and 200 μM bromsulphthalein for OAT4). Data on the *y*-axis are the rates of uptake of the prototypic substrate normalized to protein concentration. Data are from three independent experiments, each conducted in triplicate. Technical variability within each independent experiment is shown in Appendix A. Statistically significant differences in the rate of uptake of the prototypic substrate in the presence and the absence of the inhibitor (marked in red) were determined using the paired *t*-test. * *p* < 0.05; ** *p* < 0.01.

**Figure 4 ijms-25-12036-f004:**
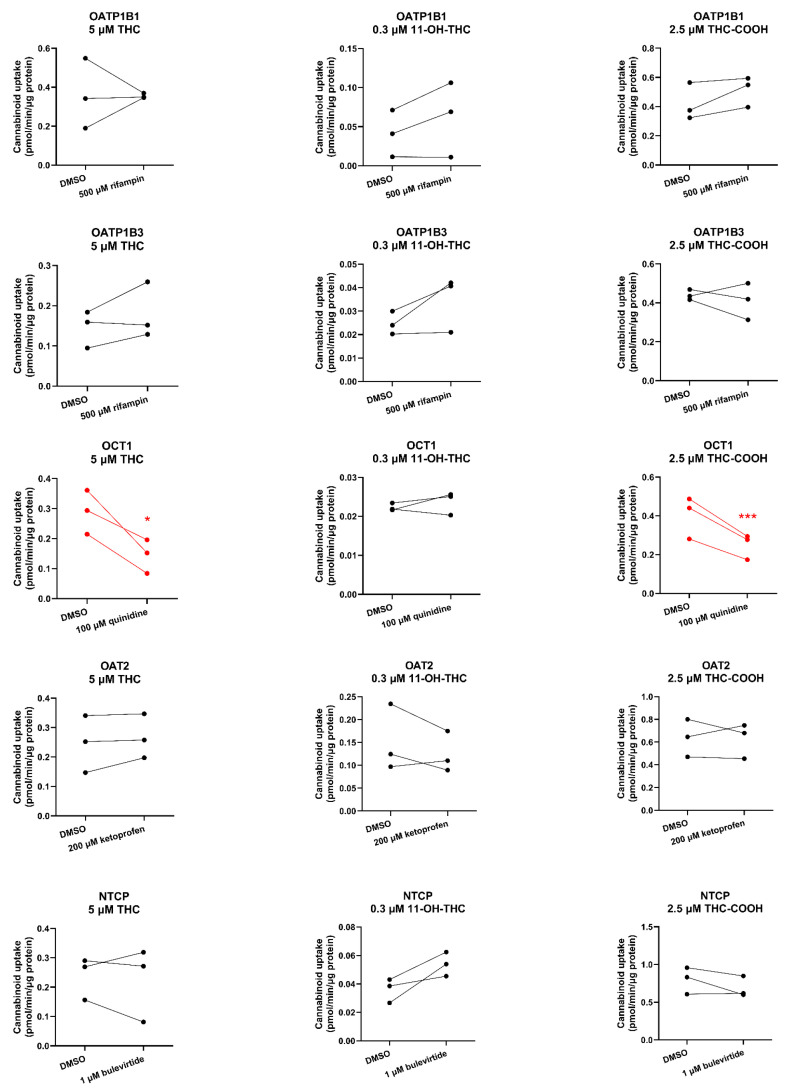
Cannabinoids as substrates of hepatic transporters. Uptake (over 15 s) of 5 μM THC, 0.3 μM 11-OH-THC, or 2.5 μM THC-COOH by human sinusoidal hepatic transporters in the absence or presence of their respective inhibitors (500 μM rifampin for OATP1B1/1B3, 100 μM quinidine for OCT1, 200 μM ketoprofen for OAT2, and 1 μM buleviritde for NTCP). Data on the *y*-axis are the uptake rate of the cannabinoid normalized to protein. Data are from three independent experiments, each conducted in triplicate. Technical variability within each independent experiment is shown in Appendix A. Statistically significant differences in cannabinoid uptake in the presence or absence of the inhibitor (marked in red) were determined using the paired *t*-test. * *p* < 0.05; *** *p* < 0.001.

**Figure 5 ijms-25-12036-f005:**
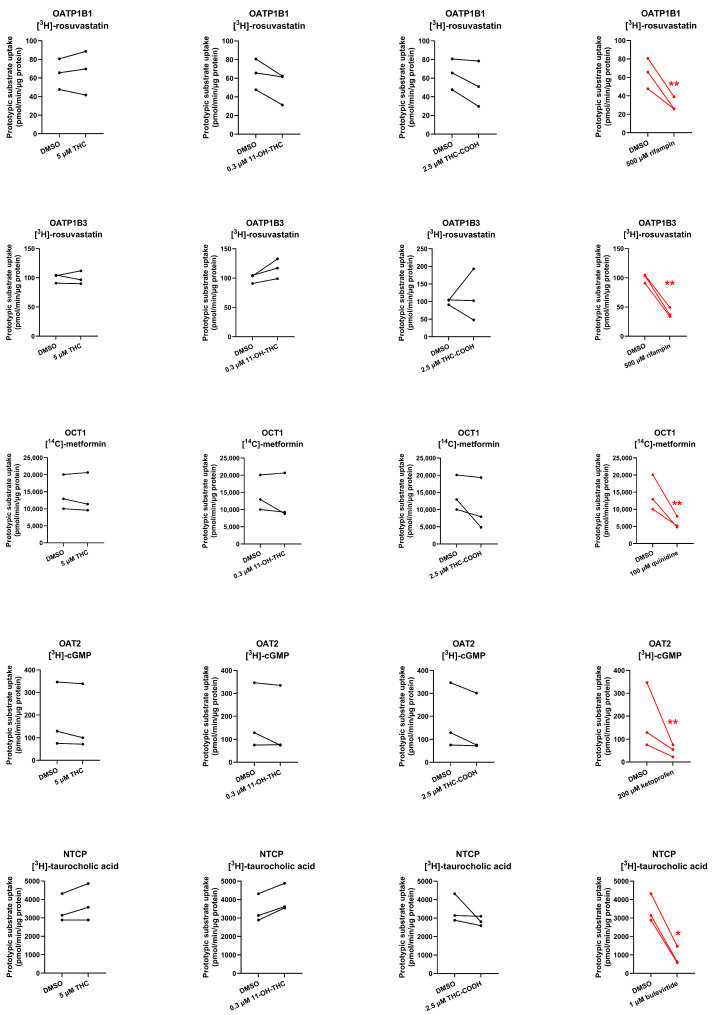
Cannabinoids as inhibitors of hepatic transporters. Uptake (over 2 min for OATP1B1/OATP1B3/OAT2/NTCP and 10 min for OCT1) of the prototypic substrates of the hepatic transporters OATP1B1/1B3 ([^3^H]-rosuvastatin), OCT1 ([^14^C]-metformin), OAT2 ([^3^H]-cGMP), and NTCP ([^3^H]-taurocholic acid) in the presence or absence of 5 μM THC, 0.3 μM 11-OH-THC, 2.5 μM THC-COOH, or their prototypic inhibitors (500 μM rifampin for OATP1B1/1B3, 100 μM quinidine for OCT1, 200 μM ketoprofen for OAT4, and 1 μM buleviritde for NTCP). Data on the y-axis are the rates of the prototypic substrate uptake normalized to protein concentration. Data are from three independent experiments, each conducted in triplicate. Technical variability within each independent experiment is shown in Appendix A. Statistically significant differences in the rate of uptake of the prototypic substrate in the presence or absence of the inhibitor or cannabinoids (marked in red) were determined using the paired *t*-test. * *p* < 0.05; ** *p* < 0.01.

**Table 1 ijms-25-12036-t001:** Substrates and inhibitors used.

Transporter	Substrate	Inhibitor
OATP1B1	40 nM [^3^H]-rosuvastatin	500 μM rifampin
OATP1B3	40 nM [^3^H]-rosuvastatin	500 μM rifampin
OATP2B1	25 nM [^3^H]-estrone-3-sulfate	10 μM erlotinib
OCT1	0.91 μM [^14^C]-metformin	100 μM quinidine
OCT3	0.91 μM [^14^C]-metformin	100 μM corticosterone
OAT2	40 nM [^3^H]-cGMP	200 μM ketoprofen
OAT4	25 nM [^3^H]-estrone-3-sulfate	200 μM bromsulphthalein
NTCP	100 nM [^3^H]-taurocholic acid	1 μM bulevirtide

## Data Availability

The authors declare that all the data supporting the findings of this study are contained within the paper.

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
