# Peer review of "Are Δ9-Tetrahydrocannabinol and Its Major Metabolites Substrates or Inhibitors of Placental or Human Hepatic Drug Solute-Carrier Transporters?"

_ijms, 2024, doi:10.3390/ijms252212036_

Round 1
Reviewer 1 Report
Comments and Suggestions for Authors
In this work, the authors investigated whether the hepatic clearance of THC and THC-COOH may involve OCT1-mediated cannabinoid-drug interactions, provided the fraction transported by OCT1 in vivo is significant. Additionally, they explored whether basolateral placental transporters (OATP2B1, OCT3, and OAT4) could explain the reduced fetal exposure to THC in humans.
The manuscript was prepared appropriately and contains no typos or incorrect sentences.
It is surprising that the materials and methods section is placed after the discussion. While the logical structure is still understandable, it might be better to switch these two sections.
Including more figures in the manuscript would have been helpful for enhancing comprehension. Currently, there is only one figure aside from those presenting specific results. Creating a summary figure that encapsulates the findings would be worthwhile.
The reference list seems sparse, though the essential citations are present in the introduction and discussion. It might have been useful to explore the background and the epidemiology of THC use in greater depth.
Author Response
Comments 1: It is surprising that the materials and methods section is placed after the discussion. While the logical structure is still understandable, it might be better to switch these two sections.
Response 1: Thank you for your comment regarding the placement of the Materials and Methods section in our manuscript. However, we would like to clarify that this structure adheres to the journal's specific format requirements, and the journal provides a template for manuscript preparation, which we followed closely to ensure compliance.
Comments 2: Including more figures in the manuscript would have been helpful for enhancing comprehension. Currently, there is only one figure aside from those presenting specific results. Creating a summary figure that encapsulates the findings would be worthwhile.
Response 2: We appreciate your suggestion to include a summary figure encapsulating our findings. However, given that our research predominantly yielded negative results, with only a few exceptions, creating a comprehensive and meaningful visual representation poses significant challenges.
Comments 3: The reference list seems sparse, though the essential citations are present in the introduction and discussion. It might have been useful to explore the background and the epidemiology of THC use in greater depth.
Response 3: Thank you for your suggestion regarding the expansion of THC use epidemiology background information. While we acknowledge the potential value of a broader epidemiological context, our manuscript's primary focus is on THC use specifically during pregnancy. We believe that the epidemiological data in the pregnant population aligns more closely with the core objectives of our study.

Reviewer 2 Report
Comments and Suggestions for Authors
The article investigates the interactions of Δ9-tetrahydrocannabinol (THC) and its major metabolites with placental and hepatic solute carrier transporters. The study aims to understand whether THC and its metabolites are substrates or inhibitors of specific transporters, which is particularly relevant for evaluating fetal exposure to cannabis compounds during pregnancy.
Terms like "fetal-to-maternal" versus "maternal-fetal" should be used consistently across the text to avoid confusion.
The finding that THC-COOH is a substrate of OCT3 is interesting but ambiguous in terms of physiological relevance; adding a discussion point on its in vivo implications could be interesting.
There are missing citation markers in a few areas (such as “Table S3” or “Table S4”).
Given that certain transporters were found to be unaffected by THC, might there be non-transport mechanisms influencing THC efflux in the placenta?
Have you considered in vivo validation to support the in vitro findings, especially concerning transporter expressions and functions in the placental and hepatic tissue?
Figures 2-5 in color could indeed enhance visual engagement and make it easier for readers to distinguish between different data sets.
Expanding the conclusion would provide a more comprehensive summary of the study’s findings.
Author Response
Comments 1: Terms like "fetal-to-maternal" versus "maternal-fetal" should be used consistently across the text to avoid confusion.
Response 1: Thank you for bringing this to our attention. We have revised the manuscript to ensure consistent use of terms throughout.
Comments 2: The finding that THC-COOH is a substrate of OCT3 is interesting but ambiguous in terms of physiological relevance; adding a discussion point on its in vivo implications could be interesting.
Response 2: OCT3 is highly expressed in the human placenta but not in the human liver or kidney where OCT1 and OCT2 are respectively highly expressed (PMID: 33114309). Thus, it is likely to play a lesser role than OCT1 in the distribution of THC-COOH into the liver. Prompted by our findings, it would be interesting to determine if THC-COOH is a substrate of OCT2.
Comments 3: There are missing citation markers in a few areas (such as “Table S3” or “Table S4”).
Response 3: Thank you for your comment regarding Table S3 and S4. We have carefully reviewed our manuscript and can confirm that references to these supplementary tables have been included in all appropriate and necessary sections. If we have missed citing the supplemental tables, please let us know where and we will address the omission.
Comments 4: Given that certain transporters were found to be unaffected by THC, might there be non-transport mechanisms influencing THC efflux in the placenta?
Response 4: Yes, the most likely mechanism (other than via a transporter) is passive diffusion of THC. THC is a highly lipophilic drug with Log P of 6.97 and therefore capable of partitioning into plasma membranes. There is also evidence that fatty acid binding proteins (FABPs) may be involved in the movement of THC through a cell. We are currently investigating this mechanism.
Comments 5: Have you considered in vivo validation to support the in vitro findings, especially concerning transporter expressions and functions in the placental and hepatic tissue?
Response 5: Indeed, we have. As indicated in the introduction (lines 36-38), in vivo evidence in both humans and non-human primates suggest that THC is transported in the fetal-to-maternal direction. Also, as indicated (lines 55-60) in our most recent paper (accepted by Nature Communications but not yet in press), our PBPK model suggests that this transporter is localized to the basal membrane. Through quantitative targeted proteomics we have quantified the expression of the transporters studied in both the human placenta (of different gestational ages; reference 12) and in human livers (PMID 25534768, 24122874).
Comments 6: Figures 2-5 in color could indeed enhance visual engagement and make it easier for readers to distinguish between different data sets.
Response 6: Thank you for this suggestion. We have updated the figures and used red color to emphasize where the transporter was significantly involved or affected.
Comments 7: Expanding the conclusion would provide a more comprehensive summary of the study’s findings.
Response 7: Thank you. We have included a summarizing paragraph at the end of the conclusion.
